
# Simulations of Black Carbon Over Indian Region: Improvements and Implications of Diurnality in Emissions

Gaurav Govardhan[1,2], Sreedharan Krishnakumari Satheesh[1,2], Krishnaswamy Krishna Moorthy[1], and Ravi Nanjundiah[1,2,3]

[1]Centre for Atmospheric and Oceanic Sciences, Indian Institute of Science, Bangalore, India
[2]Divecha Centre for Climate Change, Indian Institute of Science, Bangalore, India
[3]Indian Institute of Tropical Meteorology, Pune, India

**Correspondence:** Gaurav Govardhan (govardhan.gaurav@gmail.com)

**Abstract.** With a view to improving the performance of WRF-Chem over the Indian region in simulating BC (black Carbon) mass concentrations as well as its short-term variations, especially on diurnal scale, a region-specific diurnal variation scheme has been introduced in the model emissions and the performance of the modified simulations has been evaluated against high-resolution measurements carried out over 8 ARFI (Aerosol Radiative Forcing over India) network observatories spread

across India for distinct seasons; pre-monsoon (represented by May), post-monsoon (represented by October) and winter (represented by December). In addition to an overall improvement in the simulated concentrations and their temporal variations, it has also been found that the effects of prescribing diurnally varying emissions on the simulated near-surface concentrations largely depend on the boundary layer turbulence. The effects are perceived fast (within about 2–3 hours) during the evening–early morning hours when the atmospheric boundary layer is shallow and convective mixing is weak, while they are delayed,

taking as much as about 5–6 hours, during periods when the boundary layer is deep and convective mixing is strong. This information would also serve as an important input for agencies concerned with urban planning and pollution mitigation. Despite these improvements in the near-surface concentrations, the simulated columnar aerosol optical depth (AOD) still remains largely underestimated vis-a-vis the satellite retrieved products. These modifications will serve as a guideline for further model-improvement initiatives at regional scale.

# 1   Introduction

The potential of aerosols to significantly offset regional climate through scattering and absorption of solar and terrestrial radiation, and modifying cloud properties is now unequivocally accepted. This includes impacts on the large-scale climate systems such as the monsoons (Chakraborty et al., 2004; Ramanathan et al., 2005; Lau et al., 2006; Vinoj et al., 2014), formation and evolution of large-scale tropical cyclones (Hazra et al., 2013; Herbener et al., 2014; Wang et al., 2014), the

terrestrial glacial cover (Yasunari et al., 2010; Nair et al., 2013; Lau et al., 2010); apart from the irreparable damage to human health (Davidson et al., 2005; Valavanidis et al., 2008; Shiraiwa et al., 2012). All such aerosol-cloud-cryosphere-climate-human interactions are important over the south Asian region, which is known to be among the hotspots of aerosols. These interactions assume societal importance as the region is home to more than a billion of human population. Though station-based



or space-based measurements help us in enhancing our understanding about such interactions, numerical models are essential to critically analyze the different components of these interactions in isolation, as well as predicting the future scenarios and helping in policy decisions. However, applications of such models are limited by the associated shortcomings. One of the common problems among many such chemistry transport models is the unrealistic simulations of aerosol loading as well as

its temporal variations over the Indian region as has been pointed out by several recent studies (Reddy et al., 2004; Chin et al., 2009; Ganguly et al., 2009; Goto et al., 2011; Nair et al., 2012; Cherian et al., 2013; Moorthy et al., 2013; Pan et al., 2015; Kumar et al., 2015; Feng et al., 2016; Govardhan et al., 2015, 2016). Examining the performance of RegCM4 model over the South Asian region, Nair et al. (2012) have found that despite near-realistic simulation of aerosol optical depth (AOD) over remote and cleaner oceanic regions, the model significantly underestimates AOD even over the moderately polluted continental

belt of the south Asian region. The study also acknowledged that the boundary layer processes within the RegCM4 model limit its performance in simulating near-surface black carbon (BC) mass concentrations over the region, during periods when the vertical dispersion is limited by shallow boundary layer. Performing an extensive performance evaluation of simulations with in-situ measurements, Moorthy et al. (2013) have reported that inadequacies in convective boundary layer parameterization and inaccuracies in emission inventories lead largely to the poor performance of GOCART simulations over the Indian region.

Carrying out multi-model evaluation, Pan et al. (2015) identified erroneous simulations of relative humidity as one of the major causes while, Govardhan et al. (2015) examining the performance of WRF-Chem over the Indian region, have concluded that emissions of BC and boundary layer parametrization are important in BC related model underestimations. Continuing it further, Govardhan et al. (2016) quantified the underestimations in BC emissions in 5 widely used inventories. Thus, the models employed to simulate aerosol loading over the Indian region appear to be impaired by unrealistic emissions of BC. Besides

the use of such a gross underestimated emissions, it is also noted that the models used time-invariant emission strength for the anthropogenic pollutants especially neglecting the diurnal variation that would arise due to time-dependent sources (vehicular traffic or domestic activities etc.).

With a view to improving the model simulations, especially in simulating their short-term variation, we prescribed a diurnal variation scheme to the emissions employed in the WRF-Chem model and used a spatially uniform adjustment factor to account

for the underestimation in BC concentration (as revealed in earlier studies). The effects of such modifications on the simulated near-surface BC mass concentrations (henceforth referred as NSBC mass concentration) and columnar AOD are examined by comparing with in-situ observations from a network of observatories as well as with space-based measurements. Additionally, the role that diurnality in BC emissions plays in controlling the simulated near-surface mass concentrations has also been examined. The details are provided section 2 and 3, the results are discussed in section 4 and the conclusions in section 5.

## 2   Model Details

The version of the WRF-Chem model used in this study is similar to the one used in our earlier studies as described in Govardhan et al. (2015, 2016). The model has been set over the Indian domain ($55^0$E-$97^0$0E, $1^0$N-$37^0$N) and is run for a period of one representative month each in pre-monsoon (May), post-monsoon (October) and winter (December) seasons of the year 2011.





The simulation months are so chosen that they capture the distinct temporal features of the aerosol loading over the region namely the pre-monsoonal maxima, the post-monsoonal build-up (after the washout) and the winter time secondary maxima as explained in fig. 1 of Govardhan et al. (2015). The region experiences a minimum in aerosol loading during June through August due to monsoonal wet-scavenging. It has been seen in our earlier studies (Govardhan et al., 2015, 2016) that the model

simulations show a good comparison with observations when the ambient aerosol loading is less. Thus, keeping in mind such a behaviour of the model we chose to carry out the model simulations and further evaluations only for those seasons which depict relatively higher aerosol burden over the Indian region (i.e. pre-monsoon, post-monsoon and winter). The rest of the model details remain the same as that given in Govardhan et al. (2015, 2016).

**2.1    Modifications of the Emission Scheme**

The prescribed emissions of BC over the Indian region that are employed in our previous studies (Govardhan et al., 2015, 2016) have been seen to be unrealistic, though they are at par with the CMIP5 emission database (Govardhan et al., 2016). In this study, the prescribed emissions, in model, of BC (and other anthropogenic pollutants such as organic carbon (OC), $SO_2$, CO, NO and $NO_2$) associated with fossil fuel combustion activities have been modified considering the realistic diurnal

variations seen across the Indian mainland (Goyal and Krishna, 1998; Sivacoumar et al., 2001); which is represented in fig.1. It depicts two characteristic peaks; one in the morning (between 8 am to 10 am) and the other in the evening (between 6 pm to 8 pm) mostly associated with the urban traffic and domestic (cooking) activities. The prescription of a diurnal variation involves multiplying the original emission intensity, which is rather time-invariant, by a diurnally varying factor (which is called 'diurnality factor' (DF)). In doing so, it is ensured that the total emissions integrated over a day remains unchanged, only its

magnitude is varied as indicated by fig.1. However, the spatial distinctiveness of the diurnality factor, which depends on the nature and strength of the region-sources, is not considered; rather it is assumed to follow the same pattern across the entire domain following (Goyal and Krishna, 1998; Sivacoumar et al., 2001). Additionally, based on the results of Govardhan et al. (2016) regarding underestimated BC emissions across India (model bias), we further modify the BC emissions by multiplying them by a uniform Adjustment Factor (AF) of 3, on top of the prescribed diurnal variation.

Upon carrying out the aforementioned modifications in emissions, we have carried out 3 sets of simulations for one representative month of each season; May (pre-monsoon); October (post-monsoon) and December (winter) for the year 2011. The 'CTRL' simulation configuration does not have the prescription of the diurnal variation and the adjustment factor to the emissions, while the 'DIEM' simulations have only the diurnal variation in the emissions of anthropogenic pollutants prescribed over the CTRL scheme. The 'DIEM+AF' configuration has the prescription of both the diurnality factor (DF) and adjustment

factor of 3 (for bias correction) on the emissions of BC. We next have examined the effects of such modifications in emissions of pollutants in-general and BC in particular on the aerosol loading and its related climatic effects over the Indian region.





## 3 Measurement Data Used

To evaluate the performance of the model with these modifications on the simulated NSBC mass concentration, we used high-time-resolution measurements of BC mass concentrations from 8 network stations across India for the same periods, where regular BC measurements are being made under the Indian Space Research Organization's 'Aerosol Radiative Forc-

5 ing over India' (ARFI) program (Moorthy et al., 2013). The stations chosen are: Bangalore (urban-continental, $12.96^0$N, $77.58^0$E), Chennai (urban-coastal, $13.08^0$N, $80.27^0$E), Hyderabad (urban-continental, $17.37^0$N, $78.48^0$E), Trivandrum (semi-urban-coastal, $8.37^0$N, $76.9^0$ E), Anantapur (semi-urban-continental $14.68^0$N, $77.6^0$E), Varanasi (semi-urban, polluted continental, $25.28^0$N, $82.97^0$E), Ranchi (semi-urban, polluted continental, $23.35^0$N, $85.33^0$E) and Dibrugarh (semi-urban, remote continental, $27.48^0$N, $95^0$E).

## 4 Results

### 4.1 Effects of Modified Emissions on NSBC: Model Vs Observations

The simulated NSBC mass concentration in CTRL and DIEM+AF model configurations are compared with the corresponding measurements, as mentioned above, for the 3 months of simulation (fig.2 and fig.3). The monthly mean diurnal variation in the measured NSBC mass concentration (black line, fig.2) over all the stations, in-general, depict the pronounced bi-modal

behaviour; with a morning peak occurring around 7:30-8:30 local time, which arises due to the combined effects of a) enhanced emissions from the domestic and traffic sectors as discussed earlier and b) fumigation effect triggered by the increasing thermal convections that breaks the nocturnal capping inversion and brings-in particles trapped in the residual layer as has been discussed in earlier works (Stull, 2012; Nair et al., 2012; Moorthy et al., 2013; Govardhan et al., 2015). Convective mixing strengthens further into the daytime and the ABL deepens, resulting in more vertical dispersion of near-surface BC and

a consequent dilution in the NSBC mass concentration. During evening to the night-time, the traffic and domestic emissions increase again, while the reduced vertical mixing and shallow nocturnal boundary layer result in the confinement of the particles near the surface, thus contributing to the high BC concentrations (black line, fig.2). The observed reductions in NSBC mass concentration post-midnight (black line, fig.2) despite the stable atmospheric boundary layer, is attributed to the reduced nocturnal emissions.

The model in CTRL configuration (blue line, fig.2), fails to simulate the magnitude of the observed NSBC mass concentration, over all the stations, for all times of the day. These gross biases, reported earlier as due to the underestimated emissions and overestimated ventilation within the model as discussed by Govardhan et al. (2015, 2016) are clearly seen. Additionally, the model fails to capture the bi-modal diurnality; though a very weak peak is indicated at some of the stations during the morning, while the evening peaks are almost absent everywhere. However, once the diurnally varying emissions with a steady adjust-

ment factor are prescribed in the models, the simulations (DIEM+AF configuration (red line, fig.2)), successfully capture not only the pair of peaks, but also the amplitudes of NSBC mass concentration over most of the observation stations. Moreover, the troughs during afternoon and late-night hours in the measured NSBC mass concentration are also realistically captured





(in magnitude and the pattern) in the simulation. Such substantial improvements in the model simulations clearly bring-in the effect of incorporating scaled-up version of emissions with a diurnal variation (which is more realistic than constant emissions) in the simulations and emphasize its need. In a stricter sense, the diurnal variation in emissions of BC as well as the adjustment factor would vary from station to station depending upon the nature of sources and mesoscale meteorology running over the

synoptic scale (Govardhan et al., 2016). Thus, the modified emissions may not necessarily improve the model's performance homogeneously throughout the study domain. This is especially clear from disagreements that still persists between model (DIEM+AF configuration) and the observations at a few stations (fig.2d Chennai May, fig.2i. Bangalore October, and fig.2l. Hyderabad October). Nevertheless, in-general, the present scheme yields a large improvement over the CTRL simulations.

We next compare the simulated daily mean NSBC mass concentration with the corresponding observations. Since the CTRL

simulations fail to capture the magnitude and the pattern of the diurnal variation (fig.2), it significantly underestimates the observed daily mean values of NSBC mass concentration (as indicated by the blue dots in fig.3). The CTRL simulations also yield NSBC mass concentration within a much narrower range than the measurements. On the other hand, the simulations with diurnally varying emissions and adjustment factor show large improvements (red dots, fig.3). The range of the modeled NSBC mass concentration gets broadened vis-a-vis CTRL, and the model better agrees with the measurements, especially for

higher BC concentrations. Thus, in the DIEM+AF configuration, in addition to the pattern of diurnal variation, the model better captures the magnitudes daily mean mass concentrations as well, at several stations. The coefficients of a regression analysis of different model simulations with the measurements are given in table 1.

The increase in the slope of the best-fit line in the DIEM+AF vis-a-vis CTRL, could be clearly noted from columns 3 and 4 of table 1 for a large number of stations. Relatively higher changes in the slope are seen for the semi-urban stations like Trivan-

drum, Anantapur and Varanasi, while no improvement is seen over Dibrugarh and Ranchi. On the other hand, The y-intercept over all the stations shows increase in DIEM+AF configuration (column 6, table 1) vis-a-vis the CTRL configuration (column 5, table 1), especially over stations Hyderabad, Varanasi and Ranchi. Such higher values of y-intercept however indicate that the prescribed modifications in the emissions, degrade the performance of model when ambient NSBC mass concentrations are lower. Despite the improvements over most of the stations, the model still underestimates NSBC mass concentration by a

large margin over Ranchi and Dibrugarh (stations located in the eastern and north-eastern region of India respectively). This is possibly attributed to large underestimation in fossil fuel emissions of BC in this region. Moreover, Dibrugarh lies in the vicinity of the boundary of the model domain. The Burma region, which is located just east of the model boundary is known to be a home to biomass burning activities especially during the pre-monsoon season (Chan et al., 2003; Huang et al., 2016). Also there is extensive contributions due to oil refineries and brick kilns spread extensively over this region (Gogoi et al., 2017).

Considering thus, the model's performance in simulating the daily mean NSBC mass concentration gets substantially improved with the prescription of the adjustment factor and diurnally varying emissions over the previous simulations (Nair et al., 2012; Moorthy et al., 2013; Kumar et al., 2015; Govardhan et al., 2015, 2016) over the Indian region.





## 4.2 Simulated NSBC Mass Concentration: Implications of Diurnality in Emissions

In this section, we attempt to understand the effects of the diurnality in emissions on the simulated NSBC mass concentration. For this, we estimate $\Delta$BC where,

$$\Delta BC~(\%) = ((NSBC_{DIEM} - NSBC_{CTRL})/NSBC_{CTRL}) \times 100 \tag{1}$$

where, $NSBC_{DIEM}$ and $NSBC_{CTRL}$ are NSBC mass concentrations in DIEM and CTRL configurations of the model simulation respectively. Thus, $\Delta$BC signifies the effects of the prescribed diurnally varying emissions over the time-invariant emission scheme. We examine the monthly mean, 3 hourly averaged $\Delta$BC values for the month of May 2011 (as representative) in fig.4. As expected, during a day, $\Delta$BC values depict a broad range with its extremes as high as $\pm$ 40%. During midnight to morning hours (03:00 to 09:00; fig.4) $\Delta$BC shows negative tendencies roughly uniformly over the entire land-mass of the region. This signifies that the diurnal variation in emissions induces lesser ambient NSBC mass concentration during the night time than that would have been present, if the emissions had no diurnal variations. The maximum negative values ($\sim$-40%) occur during 06 hours. Similarly, in the evening to midnight sector, $\Delta$BC becomes largely positive. This indicates that, owing to the diurnality in emissions, we experience higher ambient NSBC mass concentration during evening to midnight hours as compared to the time in-variant emissions scenario. Between these two, i.e. from 09:00 to 15:00 hours (fig.4), $\Delta$BC values appear mostly close to zero. The positive changes are relatively higher over the Indo-Gangetic plains region (18:00-24:00, fig.4), while the negative changes are more predominant over the western part of India (03:00-06:00, fig.4). Thus it can be said that, the ambient diurnal variation in emissions of BC leads to cleaner mornings and mid-nights, with more polluted evenings, as compared to the time in-variant emission scenario. Such systematic effects of diurnality in emissions on NSBC mass concentration are also noticed on land-mass of Sri-Lanka; an island located near the southern tip of India. On the other hand, the ocean basins within the Indian region do not show such effects (fig.4), due to no time-dependent emissions over oceans and also due to weaker ABL dynamics. Thus, the effects of emission cycle on NSBC mass concentration occur only over the regions where time variation in BC source-strength prevails. On account of negative deviations during midnight to morning hours and positive tendencies during evening to night hours, the net effect of diurnal variation in emissions on NSBC mass concentration on daily-mean basis appear negligible.

In addition to the diurnal nature of anthropogenic emissions, the dynamics of the atmospheric boundary layer would strongly influence the near surface BC concentration, as has been shown in several studies (Nair et al., 2012; Moorthy et al., 2013; Govardhan et al., 2015). As such, we examine the relationship between $\Delta$BC and the planetary boundary layer height (PBLH) over 6 grid boxes of dimension $1^0 \times 1^0$ (fig.5), such that the grid boxes together cover the regions with higher magnitudes of $\Delta$BC. The correlation coefficients between hourly values of absolute magnitude of $\Delta$BC ($|\Delta BC|$) and the simulated PBLH in CTRL configuration ($PBLH_{CTRL}$), are computed and are listed in table 2. The consistent and significant (with $p < 0.0001$) negative correlation clearly vindicates the role of boundary layer in modulating the NSBC concentrations. This behaviour of $\Delta$BC is seen in fig.4 also, where the absolute magnitudes of $\Delta$BC are higher during the evening to early morning hours (periods when boundary layer is normally shallower) and are lower during afternoon periods when the boundary layer is deeper. While





the above coupling explains the behaviour of |ΔBC|, to understand the causes behind the sign of ΔBC (+ or -), we computed the correlation coefficients (CC) between the hourly values of ΔBC and the prescribed diurnal variation in emissions (fig.1) over the same $1^0 \times 1^0$ grid boxes (fig.5) for different values of 'lead-hour' between them (fig.6), for all the 3 months of model simulation. Here lead-k means the prescribed emission cycle leads ΔBC by 'k' hours or in other words, the effect of emission
at a given time 't' is examined on ΔBC at 't+k' hours.

It clearly emerges that ΔBC holds a strong positive relationship with the diurnal emission cycle, indicated by higher values of CC (fig.6a–c). The CC increases with increasing value of the lead hour (k); the maximum occurring for k values 3 to 4. This indicates that the effects temporal variations in the emissions are perceived the most on the near surface BC concentrations after 3 to 4 hours of the actual emissions. This is also evident from fig.4 and fig.1, where the minimum in emissions is seen to occur
at 3 am while that in ΔBC occurs at 6 am. A similar delay is also seen between maxima in emissions (18 hrs) and maxima in ΔBC (21 hours) (fig.4 and fig.1). As ΔBC is the resultant of both, boundary layer dynamics and the emissions cycle, it is the differences in the diurnality of them that would be determining the optimum 'k' value when the effect maximizes on NSBC mass concentration. This is examined further by computing the correlation coefficients between ΔBC and the emission cycle separately for the sun-lit (period when the boundary layer is dynamic) and post-sunset hours (when the boundary layer
collapses to the nocturnal layer). During the sun-lit/morning-afternoon hours (07:30-17:30 IST), when the thermal convection is significant, the CC values maximize at around lead-hour of 5-6 (fig.6d–f) (as the turbulence facilitate better dispersion of the emissions), while during the stable hours of evening tonight (18:30-06:30 IST), it maximizes within 2-3 hours (fig.6d–f) as the volume available for dispersion is smaller. It can be noticed that, ΔBC and the emission cycle are negatively correlated (CC values being negative) for the first few lead-hours of the morning time (fig.6d–f). During the morning hours, the emissions
show an increasing trend (fig.1), while ΔBC appears to be negative, due to the deepening of the boundary layer and the consequent dispersion of BC, which do not allow the increased emissions to show the same effect on concentrations immediately. A deviation from this pattern is seen during winter (December), for morning to afternoon hours (fig.6f), the regions GP2, SI and Bengal, which depict a positive CC between ΔBC and the emission cycle, and the maximum CC is seen around 3 to 5 hours, as opposed to 5—6 for the other regions (GP1, Delhi and CI) and the other seasons (fig.6d-e). Examining the simulated
PBLH over these regions, it is found that, the monthly mean daily maximum values are 804 m, 932 m and 672 m respectively, substantially lower than those over the other 3 regions (GP1-1019 m, Delhi-1214 m and CI-1923 m), showing a larger vertical confinement at the former three stations enabling NSBC mass concentration to respond to emission changes much faster than the latter three stations. Nevertheless, in-general, it can be noticed that the simulated concentrations possess some 'memory' of the emission scenario during the recent hours of stable boundary layer. This 'memory' of the simulated concentrations plays a
crucial role in causing a delayed effect of the prescribed diurnal variations in emissions on the simulated concentrations. This also explains the higher values of the CC during the evening-night sector than the morning-afternoon day time hours, as the dynamics of the PBL in the former case enables quicker response of NSBC mass concentration by enhancing the confinement by shallow PBL while in the latter case the deeper boundary layer opposes the response to the emission changes by allowing enhanced dispersion within the deeper PBL. Thus, it is evident that the response of the simulated NSBC mass concentration to
the time varying emissions is modulated by the convective mixing within the atmospheric boundary layer. This analysis also



underlines that any modification to emissions of pollutants during stable boundary layer hours of evening to morning, would not only affect the air-quality for that time-period but also would influence the air-quality scenario for the subsequent sun-lit hours of unstable boundary layer and is an important input for mitigation planning.

### 4.3 Effects of Modified BC Emissions on the Simulated AOD

While near surface BC concentration has implications for health and air quality, one of the most important aerosol parameters for regional/ global climate impact assessment is the columnar aerosol optical depth (AOD), which is the vertical integral from surface to top-of-the-atmosphere (TOA), of the extinction caused by aerosols. Despite its significance, inaccurate simulation of AOD over the Indian region has been a common problem among multiple global as well as regional chemistry transport models (Nair et al., 2012; Pan et al., 2015; Feng et al., 2016; Govardhan et al., 2015, 2016). The models tend to under/over-estimate the

regional AODs due to several factors including unrealistic emissions of anthropogenic aerosol species (Govardhan et al., 2015) and incorrect simulations of near-surface humidity (Pan et al., 2015; Feng et al., 2016). In a few previous studies (Govardhan et al., 2015, 2016), it was noticed that while WRF-Chem replicates the spatial pattern of satellite retrieved AOD fairly correctly, it underestimates the magnitudes by factors ranging from 1.5 to 2 over the Indian region. In view of effects seen with modified emission on NSBC mass concentration, we examined the effects of this on the simulated AOD within WRF-Chem, over the

Indian region. A detailed comparison of model simulated AOD in CTRL configuration with the satellite products (MODIS and MISR) has already been presented in Govardhan et al. (2016), here we mainly examine the effects of modified BC emissions on the model simulated AOD.

We compare WRF-Chem simulated AOD (CTRL and DIEM+AF configurations) with MODIS retrieved AOD (at 550 nm) for the month of May 2011, as representative. Though, the satellite retrieved AOD over land may have larger uncertainties due

to the heterogeneity in the surface reflectance (Jethva et al., 2009), we use such products in our study for model evaluation purpose, mainly due to their spatial coverage. The satellite retrieved AOD (fig.7a) shows high values (0.6-0.8) over Indo-Gangetic plains (IGP) and off the east coast of India, the causes of which are already discussed in previous studies (Govardhan et al., 2015, 2016). Over the rest of the land-mass, moderate AOD values (upto 0.6) are noticed, with a regional hotspot over central-eastern India (fig.6a). Away from the landmass, AOD gradually reduces over the water-bodies (Arabian sea and Bay

of Bengal). In general, due to prevailing winds (westerly), Bay of Bengal is seen to be under larger aerosol burden vis-a-vis the Arabian Sea (fig.7a). The model, WRF-Chem, in the CTRL configuration (fig.7b), though captures the AOD hot-spot over the IGP region and off the east coast of India, (as shown by Govardhan et al., 2016 and in fig.7b), it largely underestimates the magnitudes. In DIEM+AF configuration (fig.7c), the pattern of model simulated AOD is roughly similar to the CTRL configuration (fig.7b), however, the magnitudes have increased as expected (due to the adjustment factor). The AOD hot-spots

over IGP and east-coast of India become more intense in DIEM+AF configuration (fig.7c), while the AOD magnitudes over rest of the region (land-mass and oceans) show relatively lesser changes (fig.7c). It can well be noticed that, even in the DIEM+AF configuration, the model still underestimates AOD over the Indian region vis-a-vis satellite retrieved values. The quantification of such underestimation has been provided in table 3. The entire model domain has been divided in 10 regional boxes. The boxes are so chosen that they individually capture regions with distinct AOD features, while together they cover the entire





model domain. The names and latitude-longitude boundaries of the boxes are listed in column 1 and 2 of the table 3. It can be noticed from table 3 (column 3 and 4) that, WRF-Chem (in DIEM+AF configuration) still underestimates the MODIS retrieved AOD over the entire Indian region, by factors ranging from 1.24-2.27; being higher over the land regions (column 5, rows 2-7, table 3) than over the oceanic regions (column 5, rows 8-11, table 3). Even over the landmass, the underestimations are higher

over the regions with high AOD burden (NIGP and NW) as compared to the relatively cleaner regions (CI, SI and Bengal). Similarly, over the oceanic bodies, higher underestimations are seen over the regions which are closer to landmass (NAS and HBoB) as compared to farther oceanic regions (AS and BoB). Thus, despite the modifications in BC emissions, the simulated AOD within WRF-Chem still underestimates the satellite products. This is not surprising, as AOD is the resultant extinction of multiple aerosol species in addition to BC, and contribution of BC to AOD over the Indian region has been reported be roughly

around 11-17% (Satheesh et al., 1999; Ramanathan et al., 2001; Srivastava et al., 2011). Moreover, the diurnal variation and the prescribed multiplication factor on the BC emissions are limited to near the surface, while in the column their effects would be negligible. A further investigation is needed to understand the causes behind underestimation of AOD over the Indian region in WRF-Chem.

Notwithstanding the limited effect of BC on AOD simulations over the Indian region, we next attempt to delineate the effects

of the diurnally varying emission cycle (fig.1) and the multiplication factor on BC emissions, on the simulated AOD. For that, we compute $\Delta AOD_{cycle}$, $\Delta AOD_{AF}$ and $\Delta AOD_{cycle+AF}$ where,

$$\Delta AOD_{cycle} = AOD_{DIEM} - AOD_{CTRL} \tag{2}$$

$$\Delta AOD_{AF} = AOD_{DIEM+AF} - AOD_{DIEM} \tag{3}$$

$$\Delta AOD_{cycle+AF} = AOD_{DIEM+AF} - AOD_{CTRL} \tag{4}$$

Thus, $\Delta AOD_{cycle}$, $\Delta AOD_{AF}$ and $\Delta AOD_{cycle+AF}$, respectively represent the modification in AOD due to the emission cycle (fig.7d), the adjustment factor of 3 on BC emissions (fig.7e) and both (fig.7f). As it can be noted, the prescription of only the diurnal variation of emission induces negligible changes (5%) in the monthly mean AOD (fig.7d). Such negligible changes

in AOD are expected as the diurnal variation merely re-distributes the emissions along the vertical while maintaining the same daily accumulated value. Hence, though the emission cycle induces larger changes in the hourly values of near surface concentrations, it leaves AOD almost unperturbed. The changes in AOD due to the prescription of spatially uniform adjustment factor of 3 to the emissions of BC are seen to be relatively higher over the IGP, north-eastern India and off the east coast of India (fig.7e). The changes are mainly seen over the regions with high emission of BC and over the Bay of Bengal which gets

affected mainly due to advection from the adjoining continents. The changes in AOD (fig.7e) appear to be around 10-15% of AOD in the CTRL configuration (fig.7b). Thus, as expected a large fraction of the changes in AOD in DIEM+AF vis-a-vis CTRL configuration are due to the prescription of the adjustment factor on BC emissions (fig.7f). Thus, the simulated AOD (in



DIEM+AF configuration) remains still an underestimate over the Indian land-mass by factors more than 1.5. Possible causes behind such discrepancies could be related to the model-simulated vertical profile of aerosols, the state of mixing of aerosol in the model, which are to be examined separately.

The realistic simulations of aerosol over one of the global hot-spots, the Indian region, assume significance due to a numerous socio-economic factors. In this context, it is imperative to continuously examine and improve the performance of the models in simulating the regional aerosol loading. Learning from the previous studies (Govardhan et al., 2015, 2016), we made modifications in the WRF-Chem model in order to improve its performance in simulating the aerosol burden over the region. The subsequent large agreements between the simulated NSBC mass concentration and the station measurements over the Indian region showed in this study, on time scales as fine as an hour are noticed for the first time. Such improvements substantially enhance the applicability of WRF-Chem in characterizing the aerosol loading over the region. This study could thus serve as a guideline for further model development studies over the Indian region. It is also noted that the model still underestimates AOD (which is a common issue among many models), which could be further examined in future studies. One of the critical parameters in AOD computation is the simulated vertical distribution of aerosol. The performance of WRF-Chem in simulating altitude distribution of aerosols needs to be evaluated in this context. Another important parameter is the assumptions about state of mixing of aerosol in general and BC in particular which significantly governs the resulting optical and radiative effects of aerosol (Jacobson, 2001; Chandra et al., 2004; Peng et al., 2016). In these regards, the realistic state of mixing of aerosol over the Indian region (as revealed by a recent study (Hariram et al., 2018) could be examined and prescribed in the models to reveal its role in unrealistic simulations of AOD. This study has also revealed an important information about response of NSBC mass concentration to the changes in emissions. The advantages of reducing the emissions during the stable boundary layer hours with regards to local air-quality are well highlighted by this study. It is also noted that, the strong impact of night-time reduction in emissions on the morning-time NSBC mass concentration allows enhanced emissions during the morning hours without comprising on air-quality. This information will be vital from air-quality management perspective.

## 5 Conclusions

With a motive to improve the performance of a regional chemistry transport model WRF-Chem, in simulating near-surface BC mass concentration, the anthropogenic emissions of BC in the model are modified. The modifications include prescription of a diurnal variation and scaling-up the magnitudes by a spatially uniform factor of 3. In addition to examining the performance of the model, the role that the ambient diurnal variation in emissions plays in governing the simulated near-surface BC mass concentration, has also been analysed in this study. The main conclusions are as follows:

1. The modifications substantially improve the performance of the model in simulating near-surface BC mass concentrations giving better agreements with station measurements on the time scales as fine as an hour.




2. The effects of diurnal variation in emissions on the simulated near-surface BC mass concentration are seen mainly during the hours of shallower boundary layer height.

3. The diurnal nature of the BC emissions induces cleaner (more polluted) morning (evening to nights) hours vis-a-vis temporally independent emission scenario. However, on the daily mean basis, the effects of diurnal variation in emissions on the near-surface BC mass concentration turn out be negligible due to cancellation of positive effects during evening-night hours and negative effects during morning hours.

4. On an average, the effects of prescription of diurnal variation in emissions on the simulated near-surface BC mass concentration are perceived the maximum after 3-4 hours from emissions. The effects are noticed quicker (2-3 hours) during the hours of stable boundary layer and late (5-6 hours) during the sun-lit hours of unstable boundary layer.

5. The prescribed modifications in emissions alter the columnar AOD by ~10-15%, however it is still underestimated vis-a-vis the satellite retrievals.

*Code availability.* An open-source online regional chemistry transport model, WRF-Chem, was used to perform the simulations carried out in this study. The model is freely available at http://www2.mmm.ucar.edu/wrf/users/.

*Author contributions.* GG carried out the experiments, analysed the results and wrote the manuscript. SKS conceived the experiment, lead the discussions and fine-tuned the experiments and contributed to the manuscript. KKM contributed to the discussions and fine-tuned of the experiments, put the results in perspective and revised and improvised the manuscript. RSN participated in designing the experiment and analysis of the results.

*Competing interests.* The authors declare that they have no conflict of interest.

*Acknowledgements.* The computations for this study are carried on the computational cluster funded jointly by the Department of Science and Technology FIST program (DST-FIST), the Divecha Centre for Climate Change and the ARFI project of the Indian Space Research Organisation (ISRO). We would like to thank ARFI project for providing data regarding near-surface measurements of BC, carried out over India. This work is partially supported by MoES (grant no. MM/NERC-MoES-1/2014/002) under the South West Asian Aerosol Monsoon Interactions (SWAAMI) project. The authors would also like to thank the Computational and Information Systems Laboratory (CISL-NCAR) for the Research Data Archive.



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




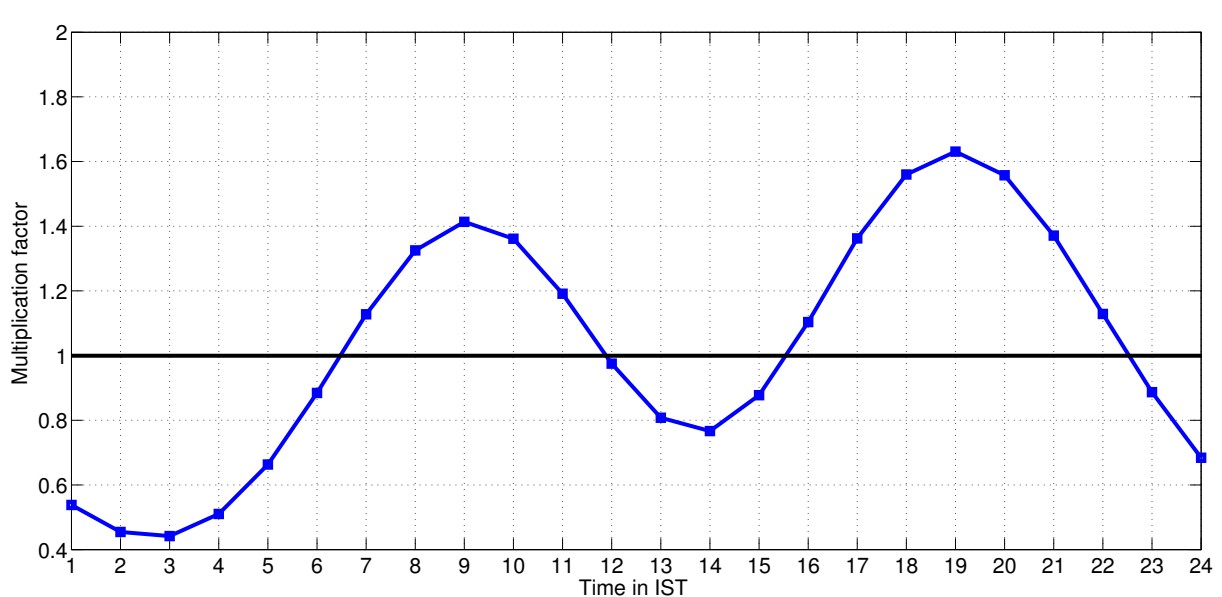

**Figure 1.** The prescribed diurnal variation to the emissions of BC,OC, Sulfate, SO$_2$, CO, NO and NO$_2$ in this study.



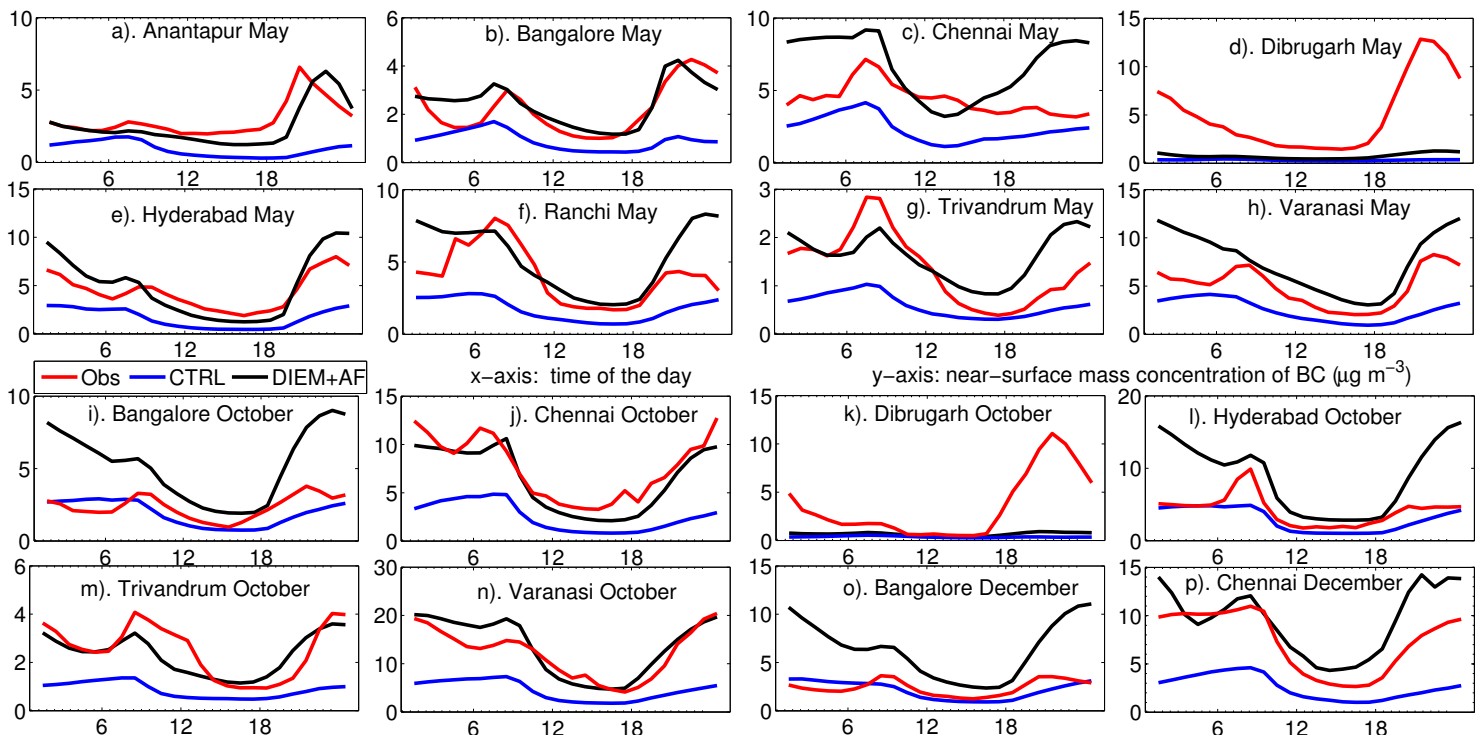

**Figure 2.** Comparison between the observed monthly mean diurnal cycle of NSBC and the model simulations in CTRL and DIEM+AF configurations, over the different ARFI observational stations across the country





**Figure 3.** Comparison between the observed daily mean NSBC and the model simulated NSBC in CTRL and DIEM+AF configurations, for the months of May 2011, October 2011 and December 2011, over the different ARFI observational stations across the country. The blue dots represent the comparison between the observations and the model simulation in CTRL configuration, while the red dots represent such comparison for the DIEM+AF configuration.





**Figure 4.** Monthly mean 3 hourly averaged values of ΔBC (%).The IST hours for each plot have been written in the bottom-right corner. The ΔBC values are plotted for May 2011 as a representative.





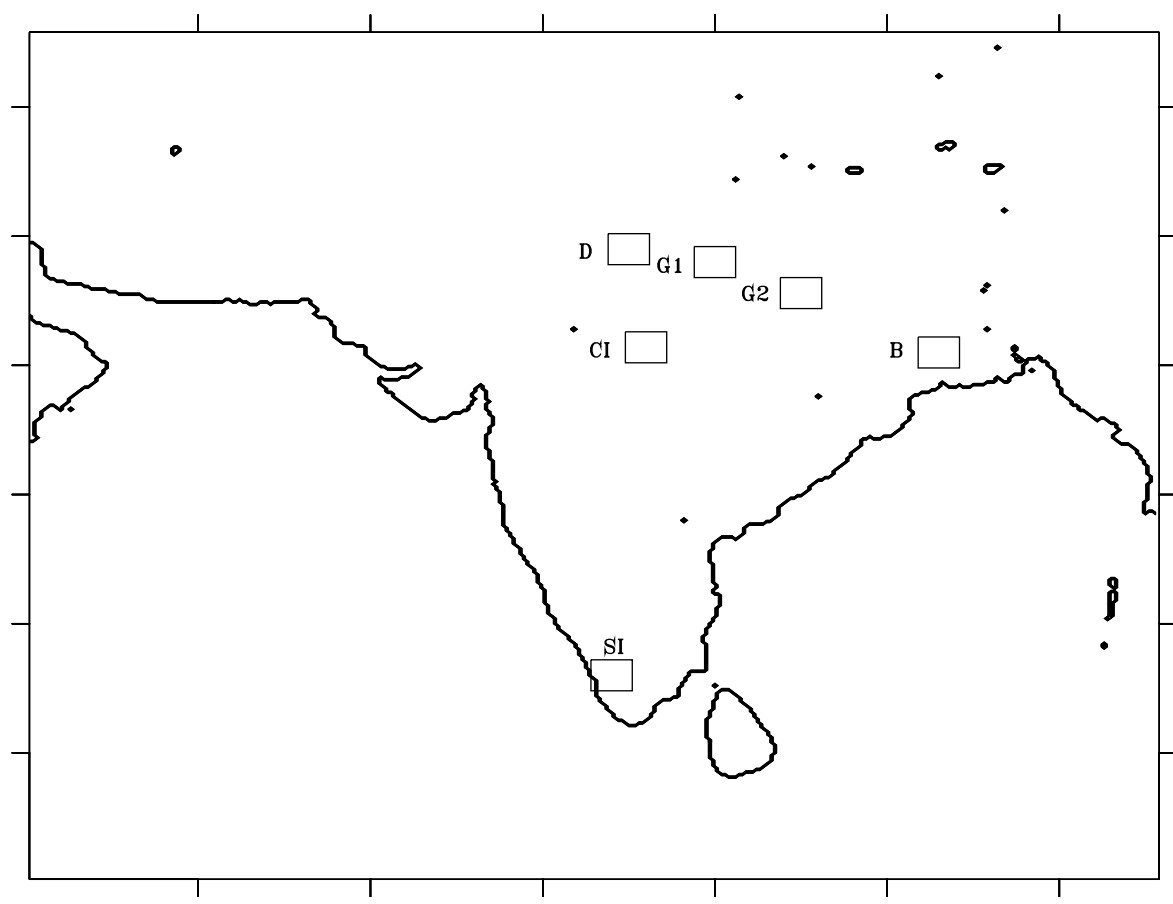

**Figure 5.** The $1^0 \times 1^0$ grid boxes chosen for the analysis of $\Delta$BC. The meaning of the acronyms are as follows; D-Delhi, G1- Gangetic plain 1, G2- Gangetic plain 2, B-Bengal, CI- Central India and SI- Southern India





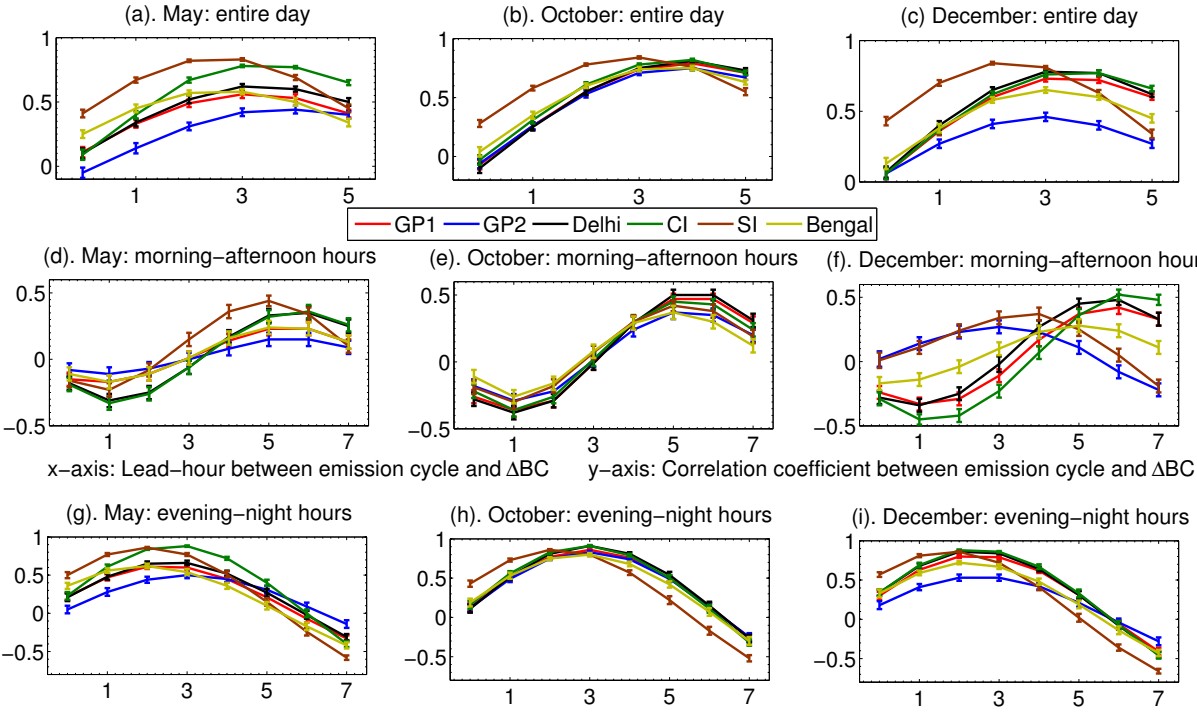

**Figure 6.** Correlation coefficients between hourly values of the emission cycle (i.e. the diurnality factor) and $\Delta$BC, at different lead-hours – considering all times of the day (00:00–24:00 IST) a) May b) October and c) December; considering only the morning–afternoon hours (07:30–17:30 IST) d) May e) October and f) December; considering only the evening–night hours (18:30–06:30 IST) d) May e) October and f) December. For a-c), d-f) and g-i) all the coefficient values with magnitude greater than 0.15, 0.2 and 0.22 respectively are significant with $p < 0.0001$.



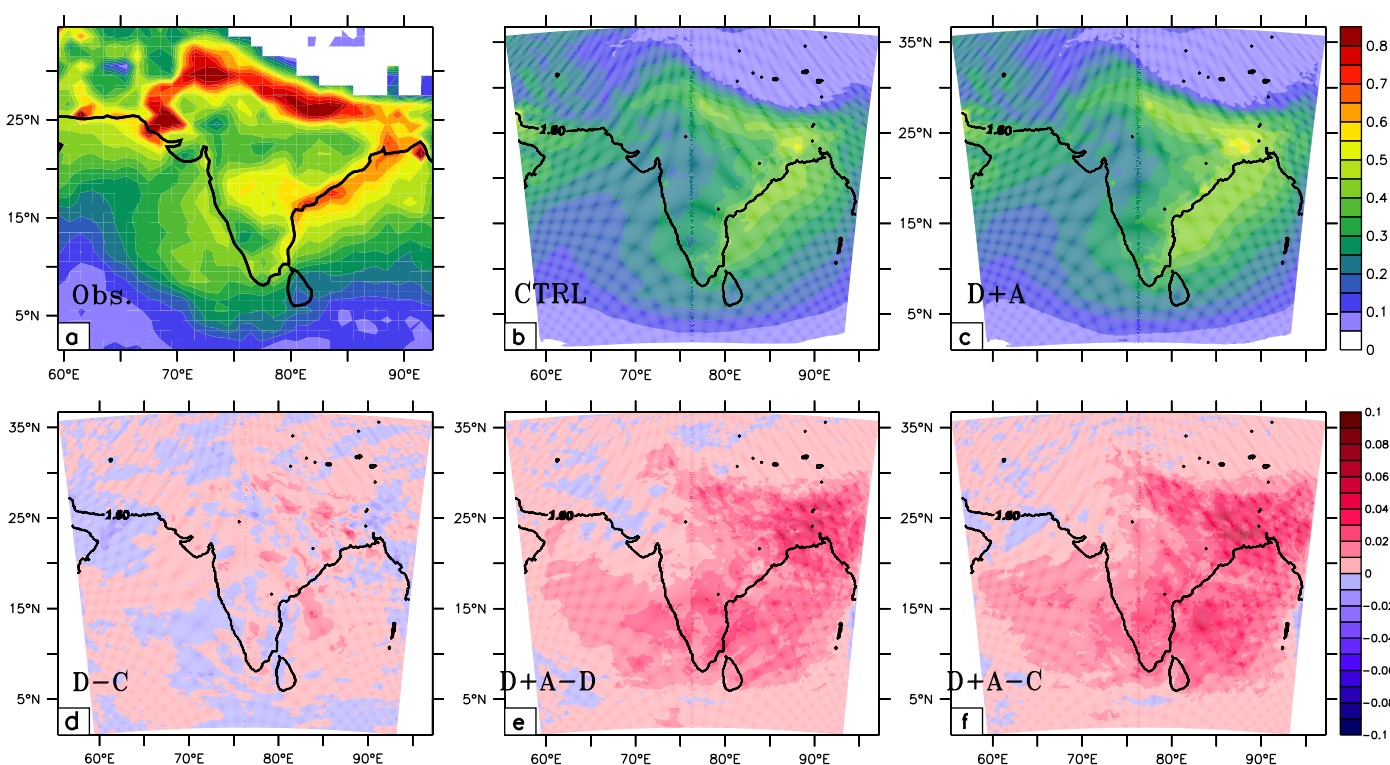

**Figure 7.** Comparison between a) satellite retrieved AOD (MODIS), b) model simulated AOD in CTRL configuration and c) model simulated AOD in DIEM+AF configuration. The bottom panels show the effects of the modifications in emissions on AOD simulations d) model simulated AOD for DIEM-CTRL configurations, which shows the effect of the prescription of the diurnal variation in emissions on AOD. e) model simulated AOD for DIEM+AF-DIEM configurations, which shows the effect of the prescription of the adjustment factor on emissions, on AOD. f) model simulated AOD for DIEM+AF-CTRL configurations, which shows the effect of the prescription of the both the diurnal variation and the adjustment factor on emissions on AOD. All the plots represent the month of May 2011.



| Station | Month | Slope | | Y-intercept | |
|---|---|---|---|---|---|
| | | CTRL | DIEM+AF | CTRL | DIEM+AF |
| Anantapur | May | 0.02 | 0.50 | 0.86 | 1.00 |
| Bangalore | May | 0.16 | 0.41 | 0.54 | 1.60 |
| | October | 0.20 | 0.59 | 1.49 | 3.92 |
| | December | 0.34 | 0.94 | 1.41 | 4.34 |
| Chennai | May | 0.24 | 0.67 | 1.30 | 3.80 |
| | October | 0.10 | 0.18 | 1.70 | 4.44 |
| | December | 0.04 | 0.36 | 1.80 | 4.27 |
| Dibrugarh | May | 0.00 | 0.00 | 0.32 | 0.76 |
| | October | 0.01 | 0.02 | 0.28 | 0.54 |
| Hyderabad | May | 0.06 | 0.14 | 1.40 | 4.30 |
| | October | 0.06 | 0.20 | 2.68 | 8.10 |
| Ranchi | May | 0.02 | 0.07 | 1.60 | 4.80 |
| Trivandrum | May | 0.06 | 0.29 | 0.51 | 1.20 |
| | October | 0.05 | 0.36 | 0.74 | 1.50 |
| Varanasi | May | 0.01 | 0.35 | 2.50 | 5.70 |
| | October | 0.12 | 0.33 | 3.01 | 9.49 |

**Table 1.** Statistics (slope of the best fit line and the Y-intercept) of the comparison between the daily-mean values of the observed and the modeled NSBC shown in fig.3.



| Region | May | October | December |
|---|---|---|---|
| Delhi | -0.50 ± 0.03 | -0.47 ± 0.03 | -0.53 ± 0.03 |
| Gangetic Plain 1 | -0.36 ± 0.03 | -0.51 ± 0.03 | -0.49 ± 0.03 |
| Gangetic Plain 2 | -0.32 ± 0.03 | -0.49 ± 0.03 | -0.46 ± 0.03 |
| Bengal | -0.38 ± 0.03 | -0.55 ± 0.03 | -0.55 ± 0.03 |
| CI | -0.51 ± 0.03 | -0.60 ± 0.02 | -0.46 ± 0.03 |
| SI | -0.55 ± 0.03 | -0.58 ± 0.02 | -0.54 ± 0.03 |

**Table 2.** The correlation coefficient between $|\Delta BC|$ and the simulated PBL height in CTRL configuration for the 6 chosen $1^0 \times 1^0$ grid boxes.



| Region | Latitude($^0$N), Longitude($^0$E) | WRF-Chem | MODIS | AF |
|--------|-----------------------------------|----------|-------|------|
| NIGP   | 73:87, 24:30      | 0.31 | 0.58 | 1.85 |
| NW     | 67:73,24:30       | 0.26 | 0.60 | 2.27 |
| CI     | 73:84,18:24       | 0.29 | 0.43 | 1.51 |
| SI     | 74:80,10:18       | 0.32 | 0.45 | 1.43 |
| LNAS   | 60:66.5,25.5:31   | 0.23 | 0.44 | 1.91 |
| Bengal | 88:92,23:28       | 0.36 | 0.48 | 1.33 |
| AS     | 58:72,6:17        | 0.19 | 0.24 | 1.32 |
| NAS    | 60:66,18:25       | 0.27 | 0.43 | 1.59 |
| BoB    | 83:94,8:16        | 0.28 | 0.34 | 1.24 |
| HboB   | 87:92,17:21       | 0.39 | 0.53 | 1.36 |

**Table 3.** Comparison of WRF-Chem simulated AOD with MODIS. AF, Adjustment Factor = MODIS AOD / Model AOD. The meanings of acronyms are as follows: NIGP: North India and Gangetic Plains, NW: North-Western part of Indian sub-continent, CI: Central India, SI: Southern India, LNAS: Landmass North of Arabian Sea, AS: Arabian Sea, NAS: Northern part of Arabian Sea, BoB: Bay of Bengal, HBoB: Head (Northern) part of Bay of Bengal