# Peer review of "Simulations of Black Carbon Over Indian Region: Improvements and Implications of Diurnality in Emissions"

_Atmospheric Chemistry and Physics, 2019_

## Referee Comment (RC1) · Anonymous Referee #1 · 17 Mar 2019

Review of the paper " Simulations of Black Carbon over Indian Region: Improvements and implications of diurnality in Emissions' by Gaurav Govardhan et al. In this paper, the authors examine the simulations of Black carbon over the Indian region using WRF-Chem model by additionally introducing diurnality in emissions from sources over the Indian region. The simulations thus made are better than the control runs. Since black carbon simulations and predictions are very important over the Indian region, this kind of sensitivity studies are very important. The previous studies have shown that the models underestimate the concentrations of black carbon over the Indian region. By introducing the diurnal variation of black carbon emissions, the authors find that the overall model simulations, especially the diurnal characteristics has improved. I rec-

ommend the present paper to be considered for publication after the authors make a revision. My detailed comments are given below: 1) I am not convinced how the scaling factor for diurnal emissions has been arrived at, that too an average factor for the whole country. Is this factor based on real observations? Why the authors did not consider the regional variations to bring better results? 2) What is the purpose of additionally doing one more simulation by multiplying a factor of 3? It is obvious that such simulations will improve the black carbon concentrations. In the conclusions, those results are not mentioned. It is an academic interest and does not add to any new knowledge. I suggest those results may be excluded from this paper. 3) In Fig 3, the scales (X and Y) are not symmetric. Put the same intervals and range and then plot a 450 line to show that red dots have improved in slope. 4) The statistical analyses (correlations and differences) always should be tested for statistical significance. 5) The study clearly brings out that there are large uncertainties in the emission inventories over the Indian region. Therefore, the future efforts should be made to improve the emission inventories of black carbon over the region. This kind of studies only are of academic interest, that too by considering one average diurnal profile of scaling factor for the whole India.

Recommendations: Revision

---

## Referee Comment (RC2) · Anonymous Referee #2 · 21 Mar 2019

The manuscript evaluates the performance of WRF-CHEM with a new diurnal emission profile added to BC emissions over India and model sensitivity to increasing the emissions by a factor of three. As has been noted in literature in several publications, the model calculated and available measurements of BC in general don't agree over India. Several solutions including scaling the emissions between factors of 3 and 5 and possible errors in the humidity profile in the WRF calculations over the sub-continent have been used to explain the differences in AOD as very few direct observations of BC or other speciated aerosols are available. This manuscript uses the observations performed under ARFI to evaluate the BC concentrations over a wide range of locations. Overall this is a well written paper that explains the methodology and results clearly and

[Figure]

I don't find any problems with the manuscript as it is. The work is not novel as there have been several previous papers treading the same path and the only new innovation here is the use of ARFI collected data which has not been used for constraining the models before.

Correction: Please correct figure 2 caption to include the description of the black, blue and red lines used in the figure.

---

## Author Comment (AC1) · 15 May 2019

We appreciate the summary evaluation on the importance of our work and the overall favorable recommendation, along with comments to be considered during revision. We have carefully considered the comments and revised the paper accordingly. Our point-by-point responses to the comments, based on which the revisions are made, are given below:

The comments from referee are written in red colored text, while the author responses are written in black colored text.

Anonymous Referee #1

1) I am not convinced how the scaling factor for diurnal emissions has been arrived at, that too an average factor for the whole country. Is this factor based on real observations? Why the authors did not consider the regional variations to bring better results?

- The characteristic shape of the proposed diurnal variation has been decided keeping in mind the daily hours of the vehicular traffic and domestic cooking activities, typically prevailing in India. The choice of the shape is mainly motivated from the guidelines provided in fig. 2 of Freitas et al. (2011), who have designed a widely used emissions pre-processor, 'prep_chem_sources', for use in the WRF-Chem model. They have suggested a double Gaussian distribution of emissions in the day, with a constraint that, the total emission of the pollutant over a day remains the same with or without the prescription of the diurnal variation.

Our prescribed diurnal variation compares well with the previous studies which have attempted to reveal diurnal variation of traffic using global activity related data (Olivier et al. (2003)), local vehicular traffic data (Pollack et al. (2006)) and concentration inversion techniques (Dutkiewicz et al. (2009)), over different regions of the world. Over the Indian region however, there are very few such studies (eg. Goyal and Krishna, 1998; Sivacoumar et al., 2001); which have attempted to derive diurnal variation of anthropogenic pollutants. Due to lack of observational data on diurnal variation of emissions over the Indian region, we resort to the guidelines outlined in Freitas et al., 2001.

We agree with the reviewer that, specification of a regionally varying diurnal variation would be more appropriate; however this would need extensive characterization of the diurnality of emissions in distinct regions/ seasons. Such studies do not exist over India, and it is hoped that our present work may provide motivation for initiating such studies. Our results in the present paper are limited by this aspect. We indicate this in the revised manuscript.

2) What is the purpose of additionally doing one more simulation by multiplying a factor of 3? It is obvious that such simulations will improve the black carbon concentrations. In the conclusions, those results are not mentioned. It is an academic interest and does not add to any new knowledge. I suggest those results may be excluded from this paper.

- The modifications carried out in the emissions of BC in the model, comprise of prescription of both the 'Diurnality Factor' (DF) and the 'Adjustment Factor' (AF=3). The improved simulations of near-surface BC are achieved only after the prescription of both the factors. The diurnality factor though controls the simulated BC on hourly time-scale, it leaves the simulated daily mean BC largely undisturbed, due to cancellation of positive changes during evening-night times and negative changes during midnight-morning hours. Thus, AF plays a crucial role in bringing the simulated BC closer to the observed BC magnitudes. Hence, we include both DF and AF in our analysis. This adjustment factor is necessitated by the in-accurate emission inventories over south Asian region and also poorer representation of atmospheric boundary layer during night-time and winter conditions when convective mixing is weak (as has been revealed by several earlier studies mentioned in the manuscript). The AF overcomes this to some extent; again with regional

differences. This aspect is mentioned in the manuscript.

3) In Fig 3, the scales (X and Y) are not symmetric. Put the same intervals and range and then plot a 450 line to show that red dots have improved in slope.
- We thankfully accept this suggestion, and the figure has been modified accordingly.

4) The statistical analyses (correlations and differences) always should be tested for statistical significance.
- We agree. The statistical significance of the correlations has been tested using the t-test by computing the p-value (Frenton and Neil, 2012). It is mentioned in the caption to fig. 6. It is also mentioned in the main text, in the revised version of the manuscript.

5) The study clearly brings out that there are large uncertainties in the emission inventories over the Indian region. Therefore, the future efforts should be made to improve the emission inventories of black carbon over the region. This kind of studies only are of academic interest, that too by considering one average diurnal profile of scaling factor for the whole India.
- We agree with the reviewer that prescription of one average profile of diurnality factor to the whole India does not represent the heterogeneity in the emission sources across the country. However, due to lack of data on diurnal variations of emissions locally, we had to resort to this option. We agree with the reviewer that in future a large efforts are needed to improve the emission inventories across India especially on the diurnal time scales. Such efforts are being undertaken very recently, though on highly regional scale and this work would bring the need for such efforts over a wider spatial scale.

References:

Dutkiewicz, V. A., Alvi, S., Ghauri, B. M., Choudhary, M. I., and Husain, L.: Black carbon aerosols in urban air in South Asia, Atmospheric Environment, 43, 1737 – 1744, doi:http://doi.org/10.1016/j.atmosenv.2008.12.043

Freitas, S. R., Longo, K. M., Alonso, M. F., Pirre, M., Marecal, V., Grell, G., Stockler, R., Mello, R. F., and Sánchez Gácita, M.: PREP-CHEM-SRC - 1.0: a preprocessor oftrace gas and aerosol emission fields for regional and global atmospheric chemistry models, Geoscientific Model Development, 4, 419–433, doi:10.5194/gmd-4-419-2011

Frenton Norman and Neil Martin, Risk Assessment and Decision Analysis with Bayesian Networks , CRC Press, 2012.

Goyal, P. and Krishna, T.: Various Methods of Emission Estimation of Vehicular Traffic in Delhi, Transportation Research Part D: Transport and Environment, 3, 309 – 317, https://doi.org/http://doi.org/10.1016/S1361-9209(98)00009-1, 1998.

Olivier, J., Jeroen, P., Claire, G., Pe´ron, Müller, and Sabine., W.: Present and future surface emissions of atmospheric compounds, POET Report no.2, EU project EVK2-1999-00011, 2003.

Pollack, A. K., Chan, L., Chandraker, P., Grant, J., Lindhjem, C., Rao, S.,Russell, J., and Tran,C.: WRAP Mobile Source Emission Inventories Update, ENVIRON International Corporation, Novato, CA. May. 2006

Sivacoumar, R., Bhanarkar, A., Goyal, S., Gadkari, S., and Aggarwal, A.: Air pollution modeling for an industrial complex and model performance evaluation, Environmental Pollution, 111, 471 – 477, doi:http://doi.org/10.1016/S0269-7491(00)00083-X, 2001.

---

## Author Response (AR1)

**Simulations of Black Carbon Over Indian Region: Improvements and Implications of Diurnality in Emissions - Reply to reviewer 1**

Gaurav Govardhan[1,2], Sreedharan Krishnakumari Satheesh[1,2], Krishnaswamy Krishna Moorthy[1], and Ravi Nanjundiah[1,2,3]

[1]Centre for Atmospheric and Oceanic Sciences, Indian Institute of Science, Bangalore, India
[2]Divecha Centre for Climate Change, Indian Institute of Science, Bangalore, India
[3]Indian Institute of Tropical Meteorology, Pune, India

**Correspondence:** Gaurav Govardhan (govardhan.gaurav@gmail.com)

We appreciate the summary evaluation on the importance of our work and the overall favorable recommendation, along with comments to be considered during revision. We have carefully considered the comments and revised the paper accordingly. Our point-by-point responses to the comments, based on which the revisions are made, are given below:

The comments from referee are written in red colored text, while the author responses are written in black colored text.

1) I am not convinced how the scaling factor for diurnal emissions has been arrived at, that too an average factor for the whole country. Is this factor based on real observations? Why the authors did not consider the regional variations to bring better results?

- The characteristic shape of the proposed diurnal variation has been decided keeping in mind the daily hours of the vehicular

10  traffic and domestic cooking activities, typically prevailing in India. The choice of the shape is mainly motivated from the guidelines provided in fig. 2 of Freitas et al. (2011), who have designed a widely used emissions pre-processor, 'prep_chem_sources', for use in the WRF-Chem model. They have suggested a double Gaussian distribution of emissions in the day, with a constraint that, the total emission of the pollutant over a day remains the same with or without the prescription of the diurnal variation.

15  Our prescribed diurnal variation compares well with the previous studies which have attempted to reveal diurnal variation of traffic using global activity related data (Olivier et al., 2003), local vehicular traffic data (Pollack et al., 2006) and concentration inversion techniques (Dutkiewicz et al., 2009), over different regions of the world. Over the Indian region however, there are very few such studies (eg. Goyal and Krishna (1998), Sivacoumar et al. (2001)); which have attempted to derive diurnal variation of anthropogenic pollutants. Due to lack of observational data on diurnal variation of emissions over the Indian region, we

20  resort to the guidelines outlined in Freitas et al. (2011).

We agree with the reviewer that, specification of a regionally varying diurnal variation would be more appropriate; however this would need extensive characterization of the diurnality of emissions in distinct regions/ seasons. Such studies do not exist over India, and it is hoped that our present work may provide motivation for initiating such studies. Our results in the present

paper are limited by this aspect. We indicate this in the revised manuscript at line 15-19, page 10.

2) What is the purpose of additionally doing one more simulation by multiplying a factor of 3? It is obvious that such simulations will improve the black carbon concentrations. In the conclusions, those results are not mentioned. It is an academic
5   interest and does not add to any new knowledge. I suggest those results may be excluded from this paper.

- The modifications carried out in the emissions of BC in the model, comprise of prescription of both the 'Diurnality Factor' (DF) and the 'Adjustment Factor' (AF=3). The improved simulations of near-surface BC are achieved only after the prescription of both the factors. The diurnality factor though controls the simulated BC on hourly time-scale, it leaves the simulated daily mean BC largely undisturbed, due to cancellation of positive changes during evening-night times and negative changes
10   during midnight-morning hours. Thus, AF plays a crucial role in bringing the simulated BC closer to the observed BC magnitudes. Hence, we include both DF and AF in our analysis. This adjustment factor is necessitated by the in-accurate emission inventories over south Asian region and also poorer representation of atmospheric boundary layer during night-time and winter conditions when convective mixing is weak (as has been revealed by several earlier studies mentioned in the manuscript). The AF overcomes this to some extent; again with regional differences. This aspect is already mentioned in the manuscript at line
15   5-9, page 5.

3) In Fig 3, the scales (X and Y) are not symmetric. Put the same intervals and range and then plot a $45^0$ line to show that red dots have improved in slope.

- We thankfully accept this suggestion, and the figure has been modified accordingly. The description has been included in the
20   modified manuscript at line 16-20, page 5.

4) The statistical analyses (correlations and differences) always should be tested for statistical significance.

- We agree. The statistical significance of the correlations has been tested using the t-test by computing the p-value (Lowry, 2014). It is mentioned in the caption to fig. 6. It is also mentioned in the main text, in the revised version of the manuscript, at
25   line 33, page 6.

5) The study clearly brings out that there are large uncertainties in the emission inventories over the Indian region. Therefore, the future efforts should be made to improve the emission inventories of black carbon over the region. This kind of studies only are of academic interest, that too by considering one average diurnal profile of scaling factor for the whole India.
30   - We agree with the reviewer that prescription of one average profile of diurnality factor to the whole India does not represent the heterogeneity in the emission sources across the country. However, due to lack of data on diurnal variations of emissions locally, we had to resort to this option. We agree with the reviewer that in future a large efforts are needed to improve the emission inventories across India especially on the diurnal time scales. Such efforts are being undertaken very recently, though on highly regional scale and this work would bring the need for such efforts over a wider spatial scale.

**Simulations of Black Carbon Over Indian Region: Improvements and Implications of Diurnality in Emissions - Reply to reviewer 2**

Gaurav Govardhan[1,2], Sreedharan Krishnakumari Satheesh[1,2], Krishnaswamy Krishna Moorthy[1], and Ravi Nanjundiah[1,2,3]

[1]Centre for Atmospheric and Oceanic Sciences, Indian Institute of Science, Bangalore, India
[2]Divecha Centre for Climate Change, Indian Institute of Science, Bangalore, India
[3]Indian Institute of Tropical Meteorology, Pune, India

**Correspondence:** Gaurav Govardhan (govardhan.gaurav@gmail.com)

We highly appreciate and thank the summary evaluation of the reviewer and the positive recommendation.

The comments from referee are written in red colored text, while the author responses are written in black colored text.

Please correct figure 2 caption to include the description of the black, blue and red lines used in the figure.

5    - We accept. The caption is corrected accordingly in the modified version of the manuscript.

[revised manuscript text omitted]